# Prediction of Flow Stress of Annealed 7075 Al Alloy in Hot Deformation Using Strain-Compensated Arrhenius and Neural Network Models

**DOI:** 10.3390/ma14205986

**Published:** 2021-10-12

**Authors:** Hongbin Yang, Hengyong Bu, Mengnie Li, Xin Lu

**Affiliations:** Faculty of Materials Science and Engineering, Kunming University of Science and Technology, Kunming 650093, China; hongbinyang2020@163.com (H.Y.); trekin@163.com (X.L.)

**Keywords:** 7075 Al alloy, flow stress, strain-compensated Arrhenius, BP-ANN

## Abstract

Hot compression experiments of annealed 7075 Al alloy were performed on TA DIL805D at different temperatures (733, 693, 653, 613 and 573 K) with different strain rates (1.0, 0.1, 0.01 and 0.001 s^−1^.) Based on experimental data, the strain-compensated Arrhenius model (SCAM) and the back-propagation artificial neural network model (BP-ANN) were constructed for the prediction of the flow stress. The predictive power of the two models was estimated by residual analysis, correlation coefficient (R) and average absolute relative error (AARE). The results reveal that the deformation parameters including strain, strain rate, and temperature have a significant effect on the flow stress of the alloy. Compared with the SCAM model, the flow stress predicted by the BP-ANN model is in better agreement with experimental values. For the BP-ANN model, the maximum residual is only 1 MPa, while it is as high as 8 MPa for the SCAM model. The R and AARE for the SCAM model are 0.9967 and 3.26%, while their values for the BP-ANN model are 0.99998 and 0.18%, respectively. All these reflect that the BP-ANN model has more accurate prediction ability than the SCAM model, which can be applied to predict the flow stress of the alloy under high temperature deformation.

## 1. Introduction

The 7000 series aluminum alloys can be strengthened by heat treatment, used for many high-end fields such as aerospace and transportation because of their high strength toughness, low density and excellent stress corrosion resistance [1,2,3]. The 7075 aluminum alloy is the earliest ultra-high strength aluminum alloy and is mainly used for the load-bearing parts of aircraft [1]. Similar to most alloys [4,5,6], the 7000 series aluminum alloys are generally produced by various thermoplastic processes, including extrusion, forging and rolling. However, their thermoplastic forming is complicated, because the deformation behavior is controlled by deformation conditions including strain, strain rate and temperature [7,8]. Therefore, investigating the hot deformation behavior of the 7075 Al alloy under different conditions is helpful to optimize process parameters for obtaining excellent microstructure and mechanical properties.

As an important means to characterize the relationship between the flow stress of materials and deformation parameters, the constitutive model is widely used, and also can provide great convenience for the study of hot deformation characteristics in finite element simulation [4]. Reliable simulation results require an accurate constitutive model, which is always pursued by scholars [9,10,11]. Many scholars have devoted themselves to establishing various constitutive models to describe the mechanical behavior of materials during hot working. Some typical models are proposed, including phenomenological model, physical model and artificial neural network (ANN) model [12,13]. Recently, in the prediction of high temperature flow stress in aluminum alloys, a phenomenological constitutive model called the strain-compensated Arrhenius model (SCAM) has been extensively employed [4,14,15,16,17,18,19,20]. Meanwhile, some statistical methods such as correlation coefficient (R) and average absolute relative error (AARE) are often employed to characterize the predictive power of the constitutive model. Rezaei Ashtiani et al. [16], predicted the flow stress of the AA1070 alloy at high temperature by the SCAM model. As the R value reached 99.2%, they believed that a considerable prediction effect had been achieved. Haghdadi et al. [17], developed a SCAM model for flow stress prediction of the A356 alloy, and obtained a satisfactory R value of 0.991 and an AARE value of 8.1%, respectively. In predicting the flow stress of the 7050 alloy, Li et al. [18], also adopted the SCAM model, and achieved good prediction results, with the R and AARE of 0.9922 and 6.28%, respectively. In addition, Gan et al. [19] and Dai et al. [20], predicted the flow stress of the 6063 and the AA5083 alloys, respectively, based on the SCAM model. In their respective studies, the AARE for the models were 4.65% and 4.52%, respectively. In summary, the SCAM model is suitable for predicting the flow stress of aluminum alloy in the process of hot deformation to a certain extent. Although the AARE obtained using the SCAM model was almost in the range of 4.5–8.5%, it can be further improved [12]. Generally, the flow behavior of the alloy is nonlinear, so the linear fitting method used in the establishment of the SCAM model is easy to produce a fitting error, and leads the constitutive model obtained to be not ideal.

In recent years, the ANN model has been developed as an artificial intelligence method to simulate brain neural information transmission and applied to predict thermal deformation flow stress of metal materials [21,22,23,24]. Compared with other models, the ANN model does not require repeated regression calculation. It relies on the characteristics of self-organization, self-learning and self-adaptation. It can effectively couple multiple variables, solve the nonlinear problem, and greatly improve the prediction accuracy [25,26], such as the prediction of flow stress of aluminum alloy. Li et al. [27], constructed the ANN constitutive model for the 6082 alloy, and proved that the model has a very satisfactory prediction accuracy by R and AARE being 0.99824 and 0.8%, respectively. Luo et al. [28], established a BP-ANN model to predict the thermal deformation behavior of the 7055 alloy, and confirmed that the model has a high prediction accuracy, with an average relative error of only 0.813%. It should be known that the AARE obtained by the artificial neural network model is almost less than 1%, which indicated that the ANN model had high prediction accuracy of the flow stress for aluminum alloys.

In our study, hot compression tests of the annealed 7075 Al alloy were carried out to study the flow behavior in different conditions (strain rates: 1.0, 0.1, 0.01, 0.001 s^−1^; temperature: 733, 693, 653, 613, 573 K). Based on the test data, the SCAM and BP-ANN models were derived to predict the flow stress of the alloy during hot deformation. Then, the predictive power of the two models for flow stress was evaluated and compared by residual analysis, correlation analysis and prediction error analysis.

## 2. Materials and Experiments

The chemical composition of the annealed 7075 Al alloy used in this study is shown in Table 1.

The cylindrical samples had a diameter of 10 mm and a height of 5.0 mm. Hot compression tests were performed on a TA DIL805D at the temperatures (733, 693, 653, 613 and 573 K) with strain rates (1.0, 0.1, 0.01 and 0.001 s^−1^.) Each sample was heated from room temperature to the target temperature at a rate of 5 K s^−1^, and then they were kept for 5 min. The compression stopped when the true strain was up to 0.8, and the samples were immediately cooled to room temperature at a rate of 50 K s^−1^. Figure 1 shows the detailed experimental scheme. The stress and strain data were obtained by the acquisition system. A total of 300 data points were selected from the hot compression test data (the value interval of strain is 0.05) for the construction of the constituted equation.

## 3. Results and Discussion

### 3.1. Flow Stress Characteristics

The results obtained under the predetermined experimental scheme (true stress-strain curves of the annealed 7075 Al alloy) are presented in Figure 2. For all the stress-strain curves, the flow stress increases rapidly in the initial stage of deformation, which indicates a strong work hardening phenomenon [9]. With the increase of strain, flow stress shows two flow characteristics: (1) At low strain rates conditions such as 0.001 and 0.01 s^−1^, it does not show obvious peak value, but maintains a steady flow. (2) At high strain rate conditions such as 0.1 and 1.0 s^−1^, the stress appears at an obvious peak value, then decreases slowly and finally reaches a stable state. This phenomenon is the result of work hardening and dynamic softening existing simultaneously in the hot deformation process, which compete with each other [9].

The flow stress level decreases with the increase of deformation temperature and increases with the increase of strain rate. To more intuitively reflect the influence of experimental variables on flow stress, Figure 3 shows the evolution of flow stress with different temperatures and strain rates at a strain of 0.4. It shows that the flow stress decreases with the increase of temperature at a constant strain rate, for example, at 0.001 s^−1^ with the temperature from 573 to 733 K, the stress value decreases from 64.42 to 21.38 MPa. The reason is that the increase of temperature means the increase of activation energy, which makes dislocation prone to slip [29]. In addition, when the deformation temperature increases, the critical shear stress required for the alloy to slip will decrease, which also easily leads to dislocation slip [29]. However, the flow stress increases with the increase of strain rate, for example, at 573 K, the stress value increases from 64.42 to 152.27 MPa when the strain rate ranges from 0.001 to 1.0 s^−1^. This is because the deformation time becomes shorter when the strain rate increases, which increases the deformation degree in unit time, and the softening process cannot be fully carried out, resulting in the increase of flow stress [30]. Although the flow stress of the alloy varies with temperature and strain rate, it is not a simple linear relationship.

### 3.2. Construction of SCAM Model

During the hot compression deformation, the flow stress of a material can be described by a constitutive equation. In the prediction of flow stress of materials, the well-known Arrhenius-type model is relatively common. In this model, the flow stress (σ) is calculated by strain rate ($\dot{\varepsilon }$) and temperature (T) [27]. The specific expression is as follows [31]:(1)$$\left\{\begin{array}{l} \dot{\varepsilon }=\mathrm{AF}(\sigma )\exp(-Q/\mathrm{RT})\\ {F(\sigma )=\sigma }^{n_{1}},\;(\alpha \sigma <0.8)\\ F(\sigma )=\exp(\beta \sigma ),\;(\alpha \sigma >1.2)\\ {F(\sigma )=[\sin h(\alpha \sigma )]}^{n},\;(\mathrm{for}\;\mathrm{all}\;\alpha \sigma ) \end{array}\right.$$
where $\dot{\varepsilon }$ and σ represent strain rate (s^−1^) and flow stress (MPa), respectively. T and Q represent temperature (K) and activation energy (J mol^−1^), The R that appears here is gas constant of 8.314 J mol^−1^ K^−1^. Material constants including α, n_1_, β, n, A, α = β/n_1_. In general, temperature and strain rate can be expressed by an exponential equation called the Zener-Hollomon parameter (Z), mathematically expressed as follows [31]:(2)$$Z=\dot{\varepsilon }\exp(Q/\mathrm{RT})$$

Thus, the constitutive equation including Z parameter can be described as:(3)$$\sigma =(1/\alpha )\ln\left\{{\left(Z/A\right)}^{1/n}+{\left[{\left(Z/A\right)}^{2/n}+1\right]}^{1/2}\right\}$$

It is not difficult to find that the original Arrhenius model neglected the effect of strain, which is detrimental to accurately describing flow stress. As mentioned in references [16,17,18,19,20], the change of strain will cause the change of material constants, so strain cannot be ignored. In this study, the constitutive model of the annealed 7075 Al alloy was constructed by the Arrhenius model with considering strain; the specific construction process is as follows.

#### 3.2.1. Determination of Material Constants

According to the experimental data, the material constants under different strains were calculated to establish a constitutive equation. Here, the strain of 0.4 was selected as a demonstration for determining material constants.

The method of solving the material constants can be found in References [16,17,18,19,20]. The relationship between σ-ln$\dot{\varepsilon }$ and lnσ-ln$\dot{\varepsilon }$ can be fitted by linear regression. The values of β and n1 can be solved by calculating the inverse of the slopes of each line in Figure 4a,b. The average is 0.0972 and 6.58, respectively. Thus, the value of α = β/n_1_ = 0.0148.

By taking the logarithm of Equation (1) gives:(4)$$\ln[\sinh(\alpha \sigma )]=\frac{\ln\dot{\varepsilon }}{n}+\frac{Q}{n\mathrm{RT}}-\frac{\ln A}{n}$$

Differentiating Equation (4), and *Q* can be described by an equation as follows:(5)$$Q=1000R{\left[\frac{\partial \ln\dot{\varepsilon }}{\partial \ln[\sin h(\alpha \sigma )]}\right]}_{T}{\left[\frac{\partial \left[\sin h(\alpha \sigma )\right]}{\partial \left(1/T\right)}\right]}_{\dot{\varepsilon }}$$

For a certain temperature, it can be obtained:(6)$$\frac{1}{n}={\left[\frac{\partial \ln[\sin h(\alpha \sigma )]}{\partial \ln\dot{\varepsilon }}\right]}_{T}$$

Figure 4c shows the linear fitting of ln[sinh(ασ)]−ln$\dot{\varepsilon }$, the slope can be calculated for each line, and their average is *n* = 4.78.

Similarly, when the strain rate is constant, it can be derived:(7)$$Q=1000\mathrm{Rn}{\left[\frac{\partial \ln[\sin h(\alpha \sigma )]}{\partial \ln(1/T)}\right]}_{\dot{\varepsilon }}$$

Figure 4d is the ln[sinh(ασ)]−1000/T plots, the *Q* value fitted as 121.45 kJ mol^−1^ at the strain of 0.4.

Combining Equations (1) and (2), and taking the logarithm of both sides, it can be given:(8)lnZ=nln[sinh(ασ)]+lnA

By fitting the relation between lnZ and ln[sinh(ασ)], the intercept of the fitting line is obtained, namely lnA, as shown in Figure 5, it equal to 18.27.

So far, all material constants at a strain of 0.4 have been obtained, as shown in Table 2. Material constants under other strains (the strains were set from 0.05 to 0.75 with an interval of 0.05) can also be obtained by using the similar method.

#### 3.2.2. The Compensation of Strain for Material Constants

In the previous section, the material constants at different true strains were obtained, and they are shown as dots in Figure 6. It can be observed that each material constant varies significantly with the strain, and the activation energy varies between 112.43 and 128.85 kJ mol^−1^. Compared with solution-treated 7075 alloy (283–287 kJ mol^−1^) [32], solution-treated 7050 alloy (256.6 kJ mol^−1^) [33], homogenized 7150 alloy (229.75 kJ mol^−1^) [34], homogenized Al–7.5Zn–1.5Mg–0.2Cu–0.2Zr alloy (164.075 kJ mol^−1^) [35], over-aged 7012 and 7075 alloy (141–162 kJ mol^−1^ and 143–156 kJ mol^−1^) [36], the deformation activation energy of the alloy in this study is lower. It is well known that the deformation activation energy is greatly affected by the composition and initial structure of the alloy. In addition, the Q-value is usually obtained by fitting linear regression methods, which also contributes to the difference. In general, the deformation activation energy obtained in this study is indeed relatively low, which means that the alloy has relatively good formability.

In previous studies, the material constant is usually assumed as a polynomial function of strain [16,17,18,19,20]. The same method was used in this study. By polynomial fitting, it is found that each material constant and strain satisfy a fourth-order polynomial relationship (Figure 6), and the equation as (9),
(9)$$Y{(\varepsilon )=C}_{0}{+C}_{1}{\varepsilon +C}_{2}\varepsilon^{2}{+C}_{3}\varepsilon^{3}{+C}_{4}\varepsilon^{4}$$
where, C_0_, C_1_, C_2_, C_3_ and C_4_ are the coefficients of Equation (9), they are listed in Table 3, respectively, ε is the true strain and *Y* represents variation of α, *n*, *Q* and lnA, respectively.

The strain-compensated Arrhenius constitutive model has been established, and Equation (3) can be written as Equation (10). According to Equations (9) and (10), the flow stress of the annealed 7075 Al alloy under various strains can be predicted.
(10)$$\sigma =\frac{1}{\alpha (\varepsilon )}\ln\left\{{\left(\frac{Z(\varepsilon )}{A(\varepsilon )}\right)}^{1/n\left(\varepsilon \right)}+{\left[{\left(\frac{Z(\varepsilon )}{A(\varepsilon )}\right)}^{2/n\left(\varepsilon \right)}+1\right]}^{1/2}\right\}$$

### 3.3. Construction of BP-ANN Model

In all artificial neural networks, back-propagation artificial neural network (BP-ANN) is the most extensively employed. The algorithm consists of two parts: feed-forward neural network trained and error back propagation [37]. The basic idea is to minimize the error between the actual output value and the expected output value of the network through the gradient descent search technology [27]. A typical BP-ANN is usually composed of three parts, namely, the input layer, the hidden layer and the output layer (see Figure 7) [38]. In our study, the input layer contains three neurons corresponding to three deformation parameters (temperature, strain rate and strain), while the output layer has only one neuron, which is flow stress. In order to get the number of neurons in the hidden layer, the trial-and-error method was used to optimize the number of neurons [39]. After several attempts, the optimal number of neurons was determined to be 20. Thus, the network structure established was 3 × 20 × 1.

The construction of the BP-ANN model was carried out by a self-programmed program on MATLAB 2017a software. Among 300 experimental data, the data under the deformation conditions of 613 K/0.01 s^−1^, 653 K/0.1 s^−1^ and 693 K/1.0 s^−1^ (3 × 15 = 45) were taken as test data, and the rest of the data (255) were set as training data for the BP-ANN model. To avoid the influence of dimensional difference of input data on training efficiency, it is necessary to normalize these parameters. It should be pointed out that since the change of strain rate is an order of magnitude, logarithmic transformation is performed before normalization. Normalization of data adopts the mapminmax algorithm in MATLAB, and the formula is as follows:(11)$$y=2\times \frac{x-x_{\min}}{x_{\max}-x_{\min}}-1$$
where *x* and *y* are the data before and after normalization, respectively.

In this BP-ANN training, the maximum number of iterations was set to 1000 times, the learning rate was 0.001, and the desired training error was set to 10^−6^. After the training was completed, the BP-ANN model could be obtained to predict the flow stress under different deformation conditions.

### 3.4. Evaluation of Prediction Effect of Two Constitutive Models

Figure 8 shows the comparison between the predicted values of the two models and the experimental values. The results reveal that the stress predicted by the SCAM model is less consistent with the test values than that predicted by the BP-ANN model. The residual results between the calculated values of the two models and the actual value are presented in Figure 9. It is clear that the residual for the SCAM model is about ±8 MPa, while it is less than ±1 MPa for the BP-ANN model.

To evaluate the quality of the established models, two important statistical indicators (R and AARE) were commonly used [26,37]. Corresponding calculation formulas are shown in Equations (12) and (13).
(12)$$R=\frac{\sum_{1}^{N}\left(\sigma_{\exp}^{i}-{\bar{\sigma }}_{\exp})(\sigma_{\mathrm{pre}}^{i}-{\bar{\sigma }}_{\mathrm{pre}}\right)}{\sqrt{\sum_{1}^{N}{\left(\sigma_{\exp}^{i}-{\bar{\sigma }}_{\exp}\right)}^{2}\sum_{1}^{N}{\left(\sigma_{\mathrm{pre}}^{i}-{\bar{\sigma }}_{\mathrm{pre}}\right)}^{2}}}$$
(13)$$\mathrm{AARE}=\frac{1}{N}\sum_{i=1}^{N}\left|\frac{\sigma_{\exp}^{i}-\sigma_{\exp}^{i}}{\sigma_{\exp}^{i}}\right|\times 100\%\;$$
where $\sigma_{\exp}^{i}$ represents the experimental results; $\sigma_{\mathrm{pre}}^{i}$ is the results calculated according to the established constitutive model; ${\bar{\sigma }}_{\exp}$ and ${\bar{\sigma }}_{\mathrm{pre}}$ are the average values of $\sigma_{\exp}^{i}$ and $\sigma_{\mathrm{pre}}^{i}$, respectively; *N* is the total number.

The correlation analysis results of calculated values based on the two models and experimental values are shown in Figure 10. It can be found that the data points predicted by BP-ANN model are more concentrated, with the R and AARE of 0.99998 and 0.18%, respectively (see Figure 10a.) For the SCAM model, R is 0.9967 and AARE is 3.26% respectively (see Figure 10b.) These data indicate that the BP-ANN model has a better precision in describing the hot deformation flow stress of annealed 7075 Al alloy.

Comparing between the SCAM model and BP-ANN model, the former uses linear fitting and polynomial fitting methods to derive the constants in the equation, but there were certain errors in each fitting process and it could not reach a more accurate state, so the prediction accuracy of the constitutive equation was low. On the other hand, the BP-ANN model predicts the flow stress through the network learning and pattern recognition method. It can deal with nonlinear problems well and has a strong adaptability and generalization ability [27]. Therefore, the established BP-ANN model has better performance and can be applied in the prediction of flow stress of the annealed 7075 Al alloy.

## 4. Conclusions

The deformation behavior of the annealed 7075 Al alloy at different compression conditions were investigated. For predicting its deformation behavior, constitutive equations based on SCAM model and BP-ANN model were established, and the prediction ability of the two models was evaluated and compared. According to the current research, several conclusions can be drawn:The flow stress of the annealed 7075 Al alloy decreases as temperature increases and strain rate decreases.The material constant (i.e., *α*, *n*, *Q* and ln*A*) in the SCAM model has a fourth-order polynomial relationship with the strain, and the activation energy varies in the range of 112.4312 and 128.8533 kJ mol^−1^.The flow stress predicted by the BP-ANN model is more consistent with the experimental values than that predicted by the SCAM model. The residual for the BP-ANN model was controlled within 1 MPa, while it is about ±8 MPa for the SCAM model.The R and AARE were obtained from the SCAM model are 0.9967 and 3.26%, respectively, while the R and AARE were superior in the BP-ANN model at 0.99998 and 0.18%, respectively, which reveals that the predicted accuracy of BP-ANN model is higher than that SCAM model.

## Figures and Tables

**Figure 1 materials-14-05986-f001:**
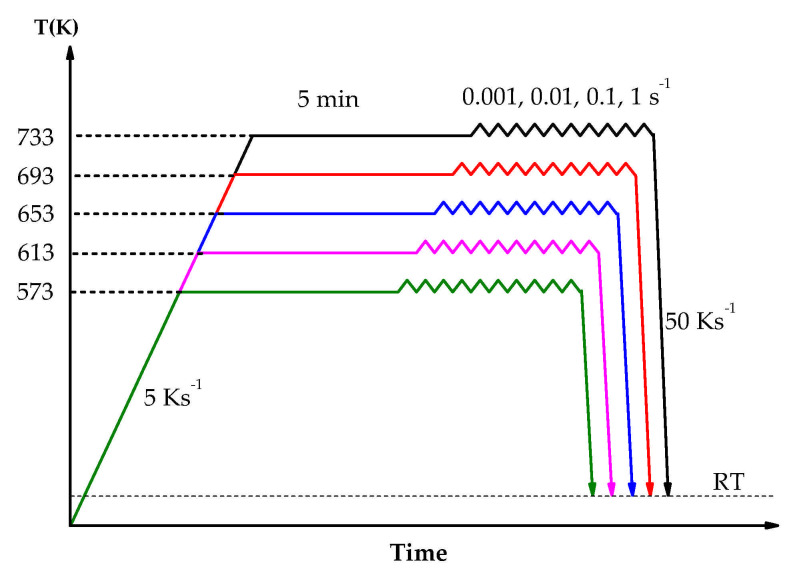
Schematic of compression tests.

**Figure 2 materials-14-05986-f002:**
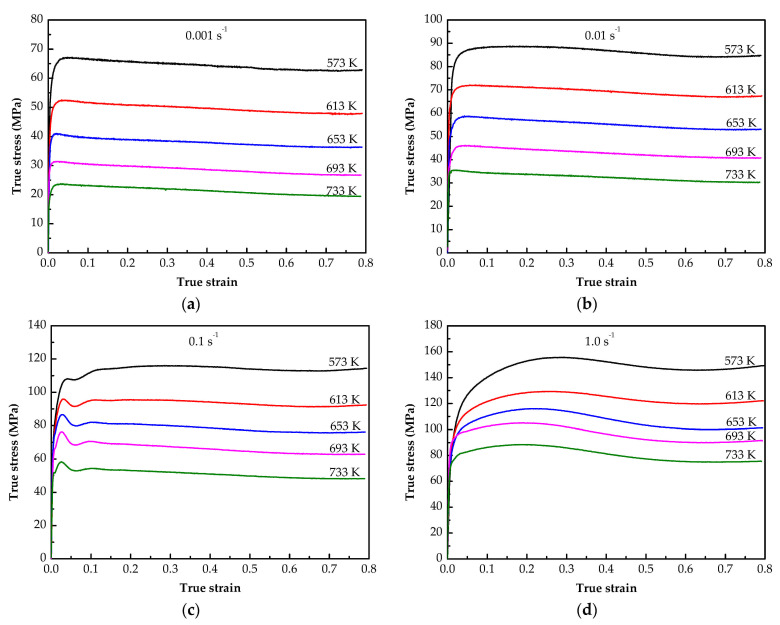
True stress-strain curves of the annealed 7075 Al alloy with various deformation conditions: (**a**) 0.001 s^−1^; (**b**) 0.01 s^−1^; (**c**) 0.1 s^−1^; (**d**) 1.0 s^−1^.

**Figure 3 materials-14-05986-f003:**
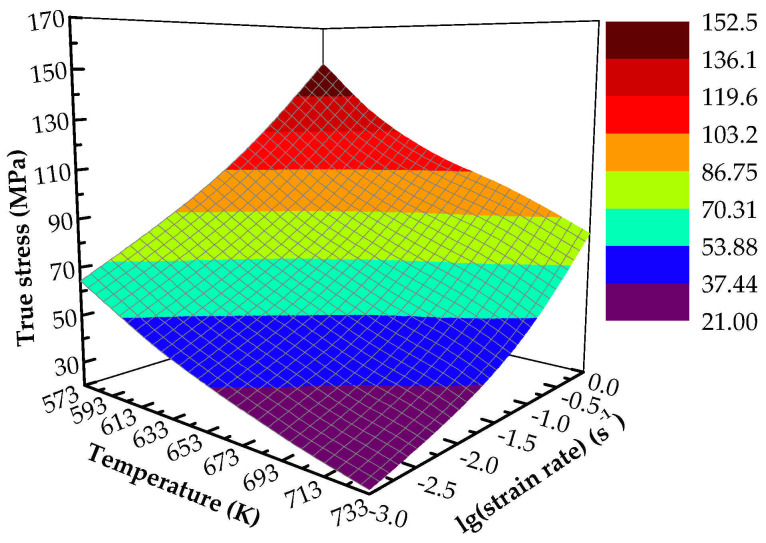
Evolution of flow stress with temperature and strain rate at true strain of 0.4.

**Figure 4 materials-14-05986-f004:**
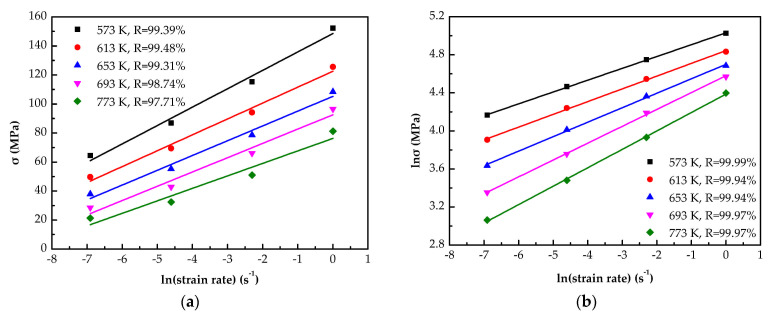
Solving material constants by linear fitting: (**a**) σ—ln$\dot{\varepsilon }$; (**b**) lnσ—ln$\dot{\varepsilon }$; (**c**) ln[sinh(ασ)]—ln$\dot{\varepsilon }$; (**d**) ln[sinh(ασ)]—1000/T.

**Figure 5 materials-14-05986-f005:**
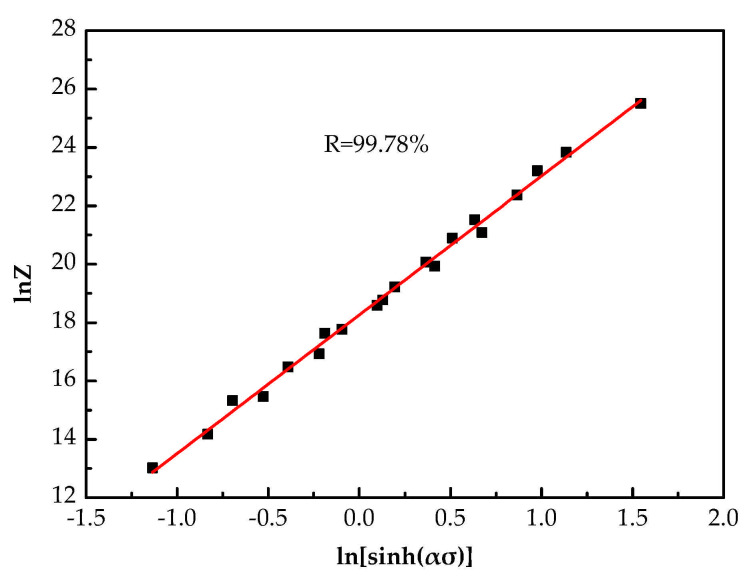
Linear relationship of lnZ—ln[sinh(ασ)].

**Figure 6 materials-14-05986-f006:**
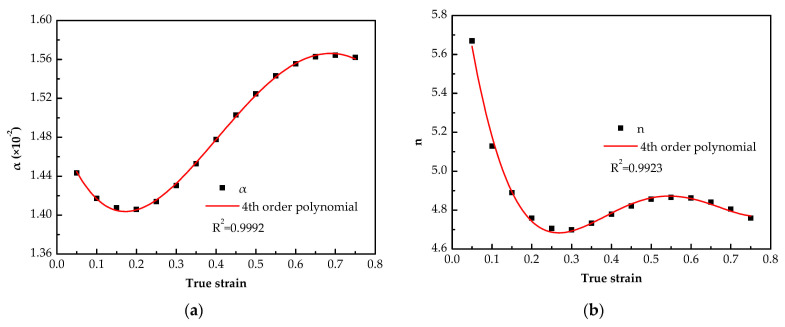
Polynomial fit between variation and true strain: (**a**) α—ε; (**b**) *n*—ε; (**c**) *Q*—ε; (**d**) lnA—ε.

**Figure 7 materials-14-05986-f007:**
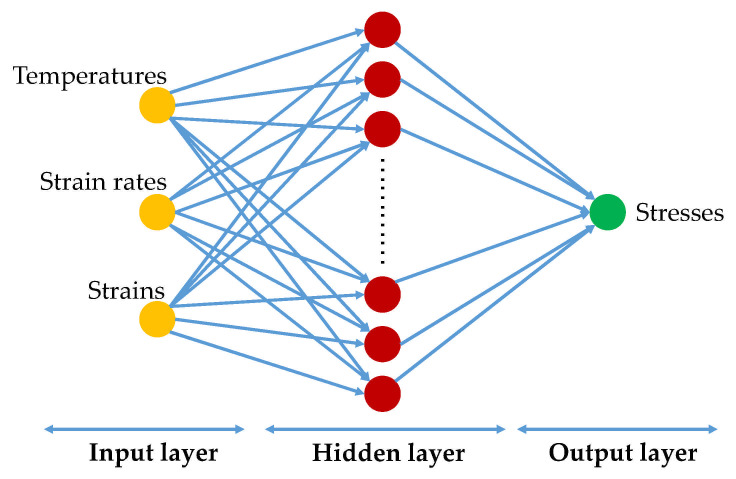
Structure diagram of BP-ANN.

**Figure 8 materials-14-05986-f008:**
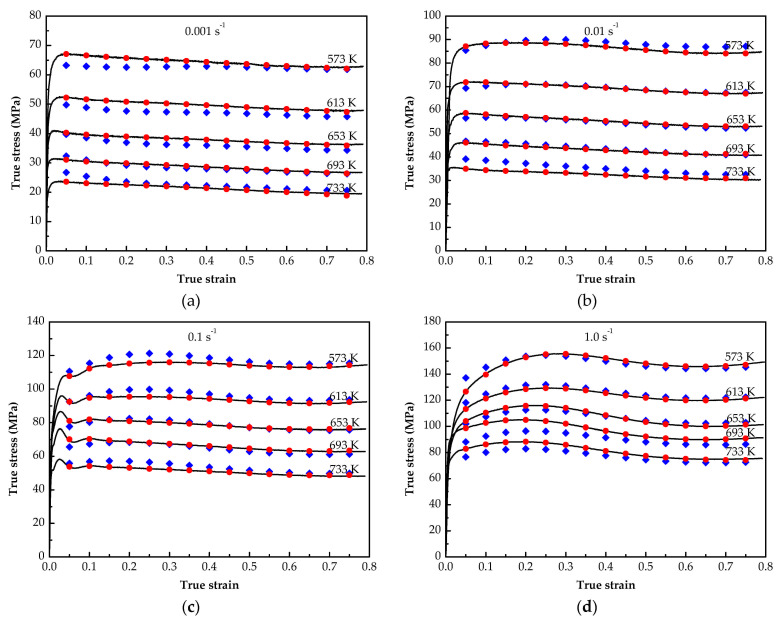
Comparison of flow stress predicted by SCAM model and BP-ANN model with experimental values: (**a**) 0.001 s^−1^; (**b**) 0.01 s^−1^; (**c**) 0.1 s^−1^; (**d**) 1.0 s^−1^.

**Figure 9 materials-14-05986-f009:**
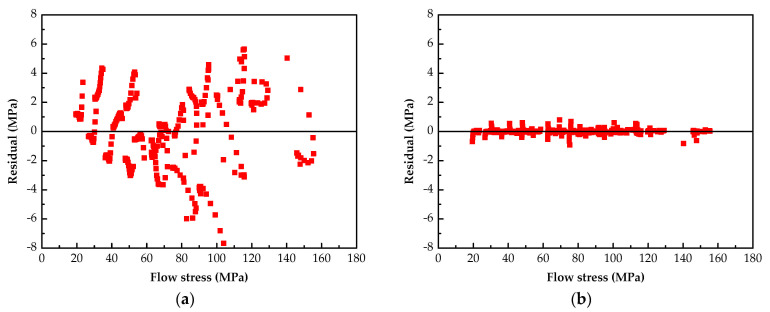
Residuals for different models: (**a**) SCAM model; (**b**) BP-ANN model.

**Figure 10 materials-14-05986-f010:**
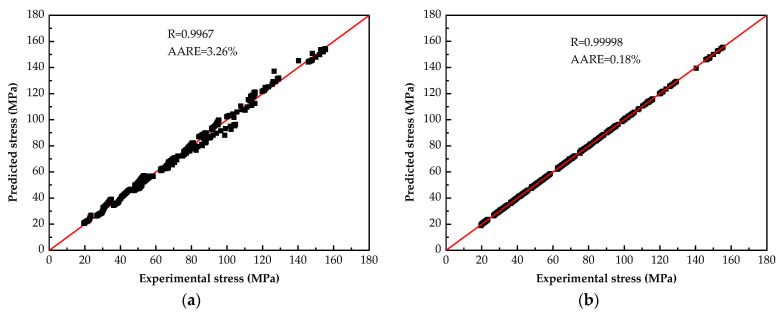
Correlation between the experimental values and predicted values: (**a**) SCAM model; (**b**) BP-ANN model.

**Table 1 materials-14-05986-t001:** Chemical composition of the annealed 7075 Al alloy (wt. %).

Fe	Si	Cr	Mn	Ti	Cu	Mg	Zn	Al
0.20	0.07	0.21	0.06	0.02	1.54	2.68	5.76	Bal.

**Table 2 materials-14-05986-t002:** Material constants at the strain of 0.4.

Parameter	α	n	Q (kJ mol^−1^)	A
Value	1.48 × 10^2^	4.78	121.45	8.5665 × 10^7^

**Table 3 materials-14-05986-t003:** The coefficients in Equation (9) for fitting α, *n*, lnA and *Q*.

Parameter	C_0_	C_1_	C_2_	C_3_	C_4_
α	1.49	−1.15	4.56	−5.02	1.54
*n*	6.33	−16.45	56.51	−77.30	36.67
lnA	19.46	−37.33	156.03	−214.29	96.95
*Q*	128.30	−214.57	896.39	−1228.97	554.15

## Data Availability

Not applicable.

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
