# Peer review of "Prediction of Flow Stress of Annealed 7075 Al Alloy in Hot Deformation Using Strain-Compensated Arrhenius and Neural Network Models"

_materials, 2021, doi:10.3390/ma14205986_

Round 1
Reviewer 1 Report
The authors of the paper "Prediction of Flow Stress of Annealed 7075 Al Alloy in Hot Deformation Using Strain-Compensated Arrhenius and Neural Network Models" have constructed two types of regression models for the flow stress and compared their predictability. It was shown that the ANN-based model has a lower error. The paper is well written and may be published after the improvement accordingly following comments:
- The friction between the sample’s edges and the dies and the adiabatic heating during the deformation may significantly influence the true stress – true strain curves [10.1016/j.jallcom.2018.08.010, 10.1134/S0031918X14080031]. Did the authors consider such influence?
- How was set and controlled the constant cooling rate of 50 K/s?
- The obtained value of the effective activation energy is too low. The authors should make a comparison with the results obtained by other scholars (e.g. 10.1007/s11665-019-04231-8, 10.1016/j.matdes.2011.05.040, etc.).
- It is recommended to use the Materials template for the manuscript.
- The number of digits in the coefficients should be decreased accordingly to the error of their values.
Author Response
Comment 1
The friction between the sample’s edges and the dies and the adiabatic heating during the deformation may significantly influence the true stress-true strain curves [10.1016/j.jallcom.2018.08.010, 10.1134/S0031918X14080031]. Did the authors consider such influence?
Response: We are thankful to the reviewer for bringing this in notice. According to the questions raised by the reviewer, we further referred to relevant literatures. Different from these literatures, the maximum strain rate used in this study is 1 s-1. Under this condition, the heat generated by deformation strain can be neglected.
Comment 2
How was set and controlled the constant cooling rate of 50 K/s?
Response: Many thanks to the reviewer for asking this question. Our hot compression tests were carried out using TA DIL805D. The samples were heated by induction and cooled using helium as cooling gas. The cooling gas was ejected from the holes of the induction coil and directly meet the surface of the sample, meanwhile the cooling rate was controlled by adjusting the gas intensity automatically. As we know, helium is one of the best cooling media, its cooling capacity is much higher than nitrogen or argon. According to the equipment manual, the cooling rate can reach 100K/s if the room-temperature helium was cooled through a liquid-nitrogen heat exchanger before spraying.
Comment 3
The obtained value of the effective activation energy is too low. The authors should make a comparison with the results obtained by other scholars (e.g. 10.1007/s11665-019-04231-8, 10.1016/j.matdes.2011.05.040, etc.).
Response: Thanks for the reviewer's suggestion. We have made careful modifications. In Section 3.2.2, we compared the activation energy obtained in this study with the values obtained by other scholars, and analyzed the benefits achieved by low deformation activation energy. The specific content has been highlighted in the article. Meanwhile, references (No. 32-36) are also added.
Comment 4
It is recommended to use the Materials template for the manuscript.
Response: We are thankful to the reviewer for providing valuable suggestions. Materials template was found on the official website and used in the manuscript.
Comment 5
The number of digits in the coefficients should be decreased accordingly to the error of their values.
Response: we are thankful to the reviewer for this suggestion. This question mainly focuses on the values obtained in Section 3.2, especially in Table 2 and Table 3. We have made careful modifications, specifically to keep the value to two decimal places, and we have highlighted the modified value.
Reviewer 2 Report
- Please mention the scale of strain rate (linear or logarithmic) for input data of BP-ANN Model.
- It is a little confusing about training data and test data for BP-ANN Model. Were the training data randomly selected from total 300 data? The accuracy in Fig.10 was obtained using only 455 training data or using total 300 data? Is it possible to separate training accuracy and testing accuracy?
Author Response
Comment 1
Please mention the scale of strain rate (linear or logarithmic) for input data of BP-ANN Model.
Response: We are thankful and grateful to the reviewer for pointing out this error. Please accept our apologies for forgetting to mention the scale of strain rate. In the revised manuscript, we have clearly stated that the strain rate was a logarithmic transformation, and the changes have been highlighted in section 3.3.
Comment 2
It is a little confusing about training data and test data for BP-ANN Model. Were the training data randomly selected from total 300 data? The accuracy in Fig.10 was obtained using only 455 training data or using total 300 data? Is it possible to separate training accuracy and testing accuracy?
Response: Thank you very much for your valuable suggestions. A total of 300 data points was selected in our study. Among these 300 experimental data, the data obtained under the deformation conditions of 613 K/0.01 s-1, 653 K/0.1 s-1 and 693 K/1.0 s-1 (3×15=45) were selected as test data, and the rest were defined as training data for the BP-ANN model. In the revised manuscript, the data used for training and testing were mentioned and highlighted. The accuracy in Fig.10 was obtained using a total of 300 data. We focused on the flow stress prediction accuracy of the SCAM model and BP-ANN model in our study, mainly comparing the overall prediction effect of the two models, so the training and testing effect of the BP-ANN model was not discussed separately.
Round 2
Reviewer 1 Report
The authors have answered previous comments and made necessary corrections to the manuscript. The paper may be accepted in the current state.